# Seamless Industry 4.0 Integration: A Multilayered Cyber-Security Framework for Resilient SCADA Deployments in CPPS

Eric Wai [1,*] and C. K. M. Lee [1,2]

1   Department of Industrial and Systems Engineering, The Hong Kong Polytechnic University,
    Hong Kong, China; ckm.lee@polyu.edu.hk
2   Research Institute of Advanced Manufacturing, The Hong Kong Polytechnic University, Hong Kong, China
*   Correspondence: eric-ch.wai@connect.polyu.hk

**Featured Application: Hardening fog/edge network connections, protecting industrial IoT (IIOT), secure cyber-physical production system (CPPS) implementation.**

**Abstract:** The increased connectivity and automation capabilities of Industry 4.0 cyber-physical production systems (CPPS) create significant cyber-security vulnerabilities in supervisory control and data acquisition (SCADA) environments if robust protections are not properly implemented. Legacy industrial control systems and new IP-enabled sensors, instruments, controllers, and appliances often lack basic safeguards like encryption, rigorous access controls, and endpoint security. This exposes manufacturers to substantial risks of cyberattacks that could manipulate, disrupt, or disable critical physical assets and processes related to their production lines and facilities. This study presents a multilayered cybersecurity framework to address these challenges and harden SCADA environments by implementing granular access controls, network micro-segmentation, anomaly detection, encrypted communications, and legacy system upgrades. The multilayered defense-in-depth (DID) approach combines policies, processes, and technologies to counter emerging vulnerabilities. The methodology was implemented in an electronics manufacturing facility across access control, zoning, monitoring, and encryption scenarios. Results show security improvements, including 57.4% fewer unauthorized access events, 41.2% faster threat containment, and 79.2% fewer hacking attempts. The quantified metrics highlight the CPPS resilience and threat mitigation capabilities enabled by the securely designed SCADA architecture, which allows manufacturers to confidently pursue Industry 4.0 integration and digital transformation with minimized disruption.

**Keywords:** SCADA security; secure CPPS; Industry 4.0; multilayered DID

## 1. Introduction

Industry 4.0 represents the vision of smart, interconnected factories where cyber-physical production systems (CPPS) enable advanced capabilities through tight integration of industrial operational technologies (OT) and information technologies (IT) [1]. By connecting sensors, instruments, controllers, machines, and appliances on the shopfloor CPPS, manufacturers aim to gain substantial operational improvements, including greater automation, intelligent analytics, predictive maintenance, dynamic optimization, and real-time supply chain coordination [2]. Supervisory control and data acquisition (SCADA) systems serving as key enablers for this connectivity and data exchange play a critical role in realizing the promise of Industry 4.0 across domains like electronics, automotive, aerospace, pharmaceuticals, and oil and gas production [2]. SCADA systems allow for the centralized monitoring and control of dispersed industrial assets, providing a window into operational status and allowing the management of equipment efficiency [3]. However, increased connectivity between corporate IT environments and industrial OT also

introduces cybersecurity risks that manufacturers need to manage proactively to avoid undermining operational resilience and safety [4]. In the electronics manufacturing field, attackers gaining access through corporate IT networks could penetrate and hijack SCADA systems on the factory floor to disrupt or shut down production lines, manipulate bill of material (BoM) databases to introduce counterfeit components, steal sensitive intellectual property like proprietary design schematics, or intentionally damage manufacturing equipment [4]. Most legacy industrial control devices and systems, including programmable logic controllers (PLC), sensor arrays, automation appliances, and single-purpose embedded devices, were designed without cybersecurity considerations in mind and, hence, incorporate few protections beyond simple passwords or limited access controls [5]. For example, SCADA networks traditionally relied on isolation from external networks and proprietary protocols rather than using rigorous authentication or encryption to secure systems. Consequently, the new attack pathways created by increasing interconnectivity between traditionally isolated OT systems and corporate IT systems require multilayered defenses across policies, processes, and technologies encompassing the full environment.

The objectives of this study are to propose and validate a multilayered cybersecurity framework encompassing access controls, network segmentation, encrypted communications, anomaly detection, and legacy upgrades to secure SCADA systems and enable electronics manufacturers to safely pursue Industry 4.0 integration. At the organizational policy level, cybersecurity best practices are defined for critical procedures like user access management, patch management, vulnerability scanning, backup and recovery, and incident response [6]. Strategic and tactical policies are set to guide security across environments. At the process layer, protection technologies like firewalls, intrusion detection systems, and role-based access controls are required to align with defined policies to provide concrete safeguards [7]. Firewalls, multifactor authentication, and network monitoring provide technical enforcement for policies. And, at the technology layer, communications are secured using cryptography, while endpoints are hardened via practices like operating system security configuration, application whitelisting, and anti-malware tools on machines [8]. Encryption, certificates, and VPNs protect communications while strong passwords, removable media controls, and system hardening counter endpoint risks. Furthermore, least privilege permissions, compliance auditing, configuration management, and redundancy in safety systems help reinforce defenses throughout the environment [9].

## 2. Literature Review

### 2.1. Brief Review of SCADA System

SCADA systems connect remote field sites and production equipment to central command centers [10]. They acquire data from sensors and instruments, communicate it to servers, and display information through monitors or human–machine interfaces (HMIs) [11]. Operators or automated controls then respond by sending commands to equipment [12]. Key SCADA capabilities include data acquisition, network communications, analysis, historical trending, and user interfaces [13].

### 2.2. Functions of SCADA

SCADA systems provide numerous operational benefits for industrial organizations. Centralized monitoring through SCADA allows operators to have visibility into the status of dispersed field sites and equipment from a single management portal, helping to improve oversight and control [14]. SCADA also enables centralized control where operators can send commands and adjust remote assets directly from the control room, streamlining management and coordination. Historical trend data collected by SCADA can be analyzed to identify inefficiencies, bottlenecks, and anomalies. Operators can then make targeted adjustments to optimize performance and throughput. The flexibility of SCADA also makes it easier to scale operations or reconfigure systems when changes are needed. Unexpected issues can also be addressed rapidly through alarms and real-time data available via SCADA interfaces. Predictive maintenance is another key benefit, as sensor readings and

usage patterns collected over time through SCADA allow maintenance teams to proactively address potential equipment failures before downtime occurs.

### 2.3. Latest Developments in SCADA

SCADA systems continue to evolve with new capabilities enabled by emerging technologies. The growing utilization of cloud computing and software-defined networking provides opportunities to enhance SCADA flexibility and scalability. Research has explored migrating traditional SCADA architectures to cloud-based models for benefits like elastic resource allocation, pay-as-you-go costs, and off-site redundancy [15]. Cloud solutions also enable mobile access and remote management of dispersed SCADA assets via Internet connectivity [16]. Integration of Industrial Internet of Things (IIoT) technologies is another area of ongoing development. Wang et al. [17] have examined leveraging wireless sensor networks and edge computing platforms to modernize supervisory control functions through real-time analytics at the network edge. This helps address issues like bandwidth limitations arising from centralized cloud models. Incorporating machine learning and AI into SCADA is also an active research area, with potential applications including predictive maintenance, advanced situation awareness, and autonomous optimization.

### 2.4. Cybersecurity Risks of SCADA

Cybersecurity remains a key concern as SCADA systems adopt new capabilities. Recent research has discussed approaches like platform diversity to reduce the cyber risks of Internet-connected systems [18]. Studies have also evaluated tailored encryption schemes and adaptive authentication methods for resource-constrained SCADA [19]. As attacks grow more sophisticated, research continuously works to enhance the resiliency of modernized SCADA infrastructure against both current and emerging threats.

SCADA connectivity to corporate environments also introduces cyber risks if not protected. Most legacy SCADA devices lack modern safeguards, leaving them vulnerable to attacks that could manipulate, disable, or destroy physical assets and processes. Potential attack pathways include HMIs, data historians, communication channels, programmable logic controllers (PLCs), remote terminal units (RTUs), input/output (I/O) servers, and sensors. By exploiting these components, unauthorized users could disrupt operations through intentional faults or failures [20].

Documented attacks directly targeting SCADA systems include Stuxnet, Havex, BlackEnergy2, CRASHOVERRIDE, and TRISIS/TRITON [21]. Their goals ranged from surveillance and data theft to process disruption to equipment destruction. For example, Stuxnet specifically targeted uranium enrichment centrifuges in Iran and manipulated their operations to cause mechanical failures over time. These incidents highlight the serious cyber-physical risks that industrial operators face as SCADA environments become more interconnected with IT systems and remote access.

### 2.5. Industry 4.0 and Cybersecurity Challenges

Industry 4.0 refers to the trend of smart manufacturing through connecting cyber-physical systems, the Internet of Things, cloud computing, and cognitive computing. This enables efficiency across the production cycle via digital connectivity and data exchange. Smart sensors integrated into machinery allow for self-monitoring and optimization. End-to-end digital integration spans the full product lifecycle. Key technologies include cyber-physical production systems (CPPS) like smart machines communicating over networks, IoT sensors for data collection, cloud platforms for flexible analytics of large datasets, and advanced techniques like predictive models, diagnostics, and decision-making. The ideal aim is for higher productivity through integrated smart factories leveraging real-time digital technologies and connectivity across research, design, production, logistics, and service. Industry 4.0 integrates operational technology on factory floors with IT systems for capabilities like predictive maintenance, intelligent supply chains, and autonomous optimization [22].

As industrial organizations adopt Industry 4.0 initiatives and integrate their operational technology (OT) environments with Information Technology (IT) systems, the level of connectivity and data exchange increases substantially. More OT devices like sensors, controllers, and machines are networked and enabled for remote access and monitoring. However, this greater interconnectivity also widens the potential attack surface for cyberthreat actors. Traditionally isolated control systems suddenly face new risks as they are connected to enterprise networks and the Internet. New access points and communications channels are opened that malicious actors can attempt to exploit to penetrate industrial environments. Without proper cybersecurity safeguards, increased connectivity essentially provides more opportunities for adversaries to infiltrate OT networks and potentially access critical manufacturing or processing systems. This interconnectivity is crucial for Industry 4.0 capabilities but needs to be managed carefully to avoid enabling cyber-attacks that can disrupt operations or damage physical assets if defenses are insufficient [23].

The growth of the Industrial Internet of Things (IIoT) is leading to more industrial devices like sensors, instruments, controllers, and appliances being connected and becoming IP-enabled [24]. However, many legacy industrial devices were designed without basic security features in mind to protect these connections [25]. Integrating newer IoT sensors with older supervisory control and data acquisition (SCADA) systems creates vulnerabilities if the security is not properly addressed [26]. The increased connectivity expands the attack surface that malicious actors could exploit. Legacy industrial systems lack modern authentication, encryption, and security monitoring capabilities. As more IIoT devices are integrated, steps need to be taken to isolate and secure both the legacy and new systems to prevent adversaries from leveraging these vulnerabilities to attack the broader industrial control system. Companies need to perform risk assessments and develop comprehensive cybersecurity strategies to safeguard IIoT and legacy devices.

The convergence of IT and OT brought about by Industry 4.0 means that operational technology environments now incorporate many of the same types of common devices, protocols, and architectures used in IT for years. This includes the adoption of commercial off-the-shelf (COTS) hardware like PCs, servers, and networking equipment in industrial settings. Legacy OT systems are also being retrofitted to connect to IT networks. As a result, OT is now facing threats familiar to IT for a long time, like malware, phishing attacks, ransomware, and the remote exploitation of known vulnerabilities. These IT-oriented risks pose new challenges for industrial cybersecurity teams accustomed to more proprietary and isolated OT environments. Without applying IT-level security practices and safeguards to emerging converged environments, manufacturers leave themselves open to the full range of sophisticated attacks routinely seen on the Internet and business networks. This integration of IT, therefore, increases the complexity of OT cyber defenses [27].

### 2.6. Cyber Defense Approach

Through a multilayered defense-in-depth (DID) framework, organizations can manage escalating challenges as SCADA environments adopt emerging Industry 4.0 technologies [28]. A DID approach applies security measures at multiple layers to provide robust protection. At the policy level, guidelines establish rules for processes like access control and patch management. Best practices need to be defined for processes like access management, patching, and backup. The process layer involves technical protections such as firewalls, intrusion detection, and authentication controls. At the process layer, technologies like firewalls, intrusion detection systems, and access controls provide protection. Finally, the technology layer strengthens security at network, endpoint, and data levels through encryption, firewalls, whitelisting, and antivirus software. And, at the technical level, encryption, VPNs, and traffic-filtering safeguard communications, while endpoint security is strengthened through hardening, whitelisting, and antivirus tools. Only a comprehensive DID strategy implementing complementary controls across policy standards, operational workflows, and technical defenses can adequately counter cyber threats targeting modern operational technology environments [29].

2.6.1. Reinforcing Protection with Additional Practices

While baseline security measures are important at each layer of defense, additional complementary practices can help strengthen protection when applied thoroughly. Least privilege access restricts users and systems only to the specific permissions required for their roles, thus minimizing the potential impact of any unauthorized access. Compliance auditing ensures technical and policy controls conform to appropriate frameworks and are operating effectively. Redundancy builds in backup and failover capabilities so that safety-critical functions are not completely reliant on any single point of failure. Robust configuration management tracks changes to industrial assets and remediates vulnerabilities or weaknesses identified through patching and updates. Collectively, practices like these form multi-layer defenses that provide greater resiliency against evolving cyber threats targeting SCADA and other operational technology [30].

2.6.2. Latest Research for Risk Mitigation in SCADA

A cybersecurity program needs the capacity to counter cyber risks at all levels, including those associated with people, processes, and technical systems [31]. At the human level, practices like user access management, regular security awareness training, and monitoring help reduce risks from intentional or careless insider behavior. Strong organizational processes are also critical implementations that are required to follow standardized methodologies, have defined change controls, and include planning for incidents and auditing adherence. Technically, defenses need depth across the network, endpoint, application, and data layers. Tools provide visibility into anomalous activity while security controls enforce policies. Continual monitoring and maintenance strengthen resilience over time. Through a balanced, comprehensive approach addressing people, processes, and technology, manufacturers can manage cybersecurity challenges in an integrated manner as widespread connectivity transforms the risk landscape. Table 1 summarizes the latest research outputs that can be applied at each layer of the DID model to mitigate risks introduced by Industry 4.0 integration and increased connectivity between IT and OT systems. The compilation highlights research focused on countering emerging vulnerabilities through comprehensive cybersecurity policies, processes, and technical controls.

**Table 1.** Summary of latest research direction for risk mitigation in 4 layers.

| Layer in DID Security Model | New Cyber Risk in SCADA | Latest Research Direction for Risk Mitigation |
|---|---|---|
| **People** | • Insider threats (intentional behavior) <br> • Human errors (careless behavior) <br> • Lack of security awareness | • User access management <br> • Monitoring and activity logging <br> • Regular security awareness training <br> • Least privilege permissions |
| **Policy** | • New risks from increased OT-IT connectivity <br> • Lack of defined strategic and tactical security policies | • Define access control policies <br> • Establish patch management best practices <br> • Set backup and recovery policies <br> • Develop incident response plans |
| **Process** | • Vulnerabilities in workflows <br> • Weak change control processes <br> • Traditional isolation of OT no longer sufficient <br> • Risks from unprotected interconnectivity | • Deploy firewalls for network security <br> • Implement intrusion detection systems <br> • Require multifactor authentication <br> • Perform compliance auditing <br> • Build in redundancy for critical systems |
| **Technology** | • Unsecured networks <br> • Vulnerable endpoints and devices <br> • Unencrypted communications <br> • Poor application security <br> • Lack of data protections | • Combined authentication and encryption <br> • Harden endpoints via OS configurations <br> • Enable VPN connections <br> • Whitelist authorized applications <br> • Deploy antivirus and malware protection |

## 3. Materials and Methods

In addition to the latest research outputs for risk mitigation summarized in Table 1, this study further proposed an integrated security framework across 4 defense-in-depth (DID) levels to harden the SCADA connectivity in CPPS for an electronics manufacturing facility with around 3000 devices. This was achieved through the following four novel techniques.

### 3.1. People Level—Granular Access Control (GAC)

The edge-level terminal-managed workstation access through NFC Badge (staff card) validation and centralized staff qualifications. After badge scanning, the edge device requests access rights and activates power outlets upon approval. This automates user authorization based on credentials and role-based GAC to MES. A Single Board PC (SBC)—Raspberry Pi edge gateway—managed operator access to production stations through badge scanning and executed the centralized MES permissions on the shop floor. All connections were secured and intrusion-proofed. The flowchart shown in Figure 1 represents a secure SCADA-empowered operator qualification system. When an operator places their staff card on the NFC reader, the Raspberry Pi submits the staff ID along with the equipment ID to the MES for validation. If the staff member is approved to use the requested equipment and if the equipment is in service condition, the Raspberry Pi will generate a 433.92 MHz 32-bit ASK key (rolling code) and send it to a Power Relay. This action will turn on the equipment, making it ready for use. The line leader and operator will be able to see the remaining approved operation time, typically counting down from four hours.

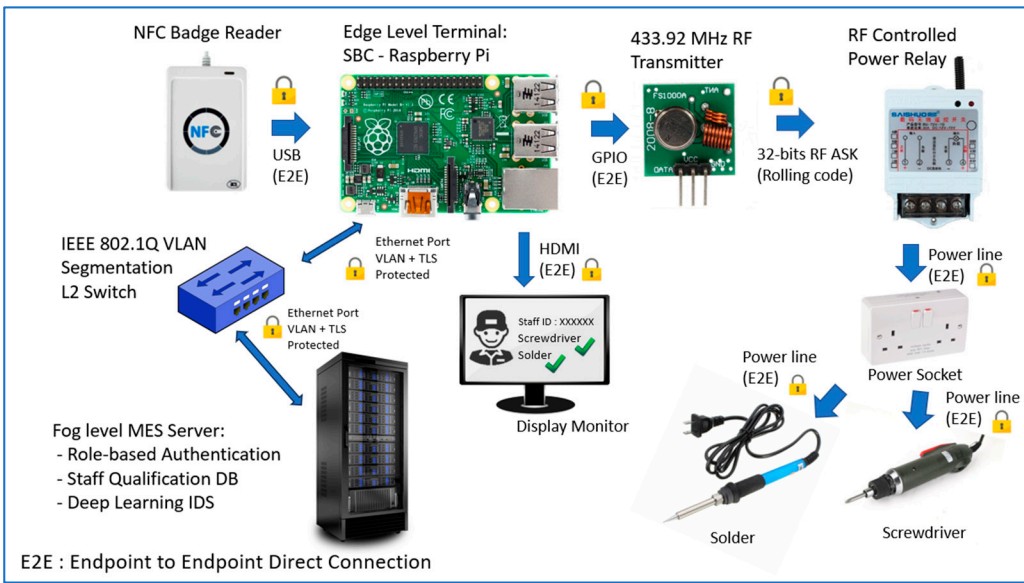

**Figure 1.** The workflow of a secure SCADA-empowered operator qualification system.

The left image of Figure 2 displays the Raspberry Pi equipped with the 433.92 MHz ASK RF Transmitter. The middle image shows this Raspberry Pi mounted on the production line and connected to an LCD monitor, demonstrating the practical integration of the secure SCADA system. The LCD monitor in the right image presents a view of the production line dashboard, providing an overview of ongoing operations and GAC users. For ethical and privacy considerations, only a minimal amount of required data will be collected in GAC.

### 3.2. Policy Level—Transport Layer Security (TLS) Protocols

Lightweight mutual authentication policy using certificates secured SCADA sessions between edge and fog layer. Sensor measurements and actuator commands were encrypted over TLS-protected links. Web dashboards leveraged HTTPS and access tokens to prevent

unauthorized access. Legacy devices were upgraded to use encrypted protocols such as rolling code cryptographic ASK (Amplitude shift keying) of insecure alternatives. Unused services were disabled to reduce the attack surface. This protected sensor and actuator data against man-in-the-middle attacks.

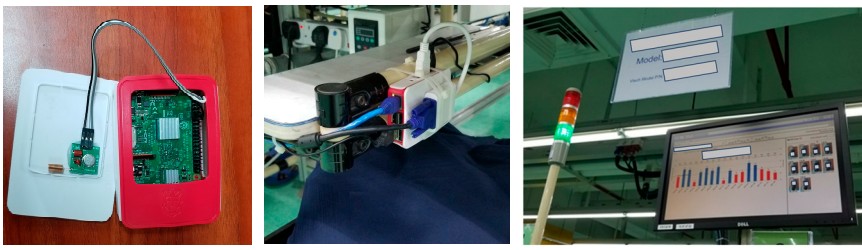

**Figure 2.** Components and user interface of SCADA system. The sensitivity information has been masked.

### 3.3. Process Level—Deep Learning Intrusion Detection Systems (IDS)

A Security Information and Event Management (SIEM) server at the fog layer utilized a deep learning LSTM model to swiftly detect network intrusion attempts by identifying anomalous traffic patterns. The intrusion detection system (IDS) generated alerts that were aggregated and correlated by the SIEM server to facilitate rapid analysis and guided response to contain identified threats. The LSTM model was trained on a dataset consisting of 30 days of normal network traffic containing 1.5 million connecting sessions. Additionally, the training data incorporated 500 examples of known cyberattack traffic, with 100 samples each of Stuxnet, Havex, BlackEnergy2, CRASHOVERRIDE, and TRISIS/TRITON traffic, to enable the model to recognize these and similar anomalous behaviors indicative of threats.

Figure 3 shows the pseudocode loads and preprocesses network traffic data, then splits it into training and test sets. An LSTM model is defined and trained to distinguish normal and anomalous traffic.

```
1   # Load network traffic data
2   data = load_network_data()
3
4   # Preprocess data
5   data = preprocess(data)
6
7   # Split data into train and test sets
8   X_train, X_test, y_train, y_test = split_data(data)
9
10  # Define LSTM model
11  model = Sequential()
12  model.add(LSTM(64, input_shape=(timesteps, features)))
13  model.add(Dense(32, activation='relu'))
14  model.add(Dense(1, activation='sigmoid'))
15
16  # Compile model
17  model.compile(loss='binary_crossentropy', optimizer='adam')
18
19  # Train model on data
20  model.fit(X_train, y_train, epochs=10, batch_size=32)
21
22  # Evaluate model on test data
23  eval_model(model, X_test, y_test)
24
25  # Use model for predictions
26  X_new = get_new_data()
27  y_pred = model.predict(X_new)
28
29  # Send alerts and containment response for anomalies
30  if y_pred < threshold:
31      send_alert()
32      block_source_ip() # contain threat
33      disable_user_account()
```

**Figure 3.** The workflow and pseudocode for monitoring the network characteristics.

The model is evaluated on the test data and then makes predictions on new traffic data. If an anomaly is detected by predictions falling below a threshold, the system triggers alerts

and automated containment actions like blocking suspicious IP addresses or disabling related user accounts. This outlines an end-to-end LSTM deep learning pipeline for network intrusion detection and containment in a concise manner.

Figure 4 shows our recent research [32] and highlights the use of recurrent neural networks (RNNs) as a deep learning (DL) technique for generating representations from raw data in applications like classification, regression, clustering, and pattern recognition. RNNs incorporate Long Short-Term Memory (LSTM) cells with input, forget, and output gates to control signal propagation. The connections between hidden layers are denoted by W, and the weight matrix U connects inputs to the hidden layer. The LSTM's unique architecture addresses the vanishing and exploding gradient problem by allowing gradients to propagate without decay through linear summation of cell states in the hidden layer. This flexibility enables the network to retain recent and past information selectively, empowering the data to determine which information is relevant. However, long-distance dependencies are difficult for standard RNNs to model effectively. LSTM networks address this issue by incorporating a memory cell that can maintain information over long periods of time. This makes LSTM well-suited for processing time series data, enabling tasks like forecasting and classification. The LSTM architecture consists of a chain of repeating modules, each containing four interacting neural layers. These layers are known as cells, and they use structures called gates to control information flow and memory updates. The recurrence weight matrix W connects the previous and current hidden states to propagate relevant context forward. The input weight matrix U transforms current inputs into a hidden representation. Candidate hidden states C are computed based on the current input and the previous hidden state. The cell's internal memory C is updated by combining the previous memory scaled by the forget gate and the new hidden state scaled by the input gate.

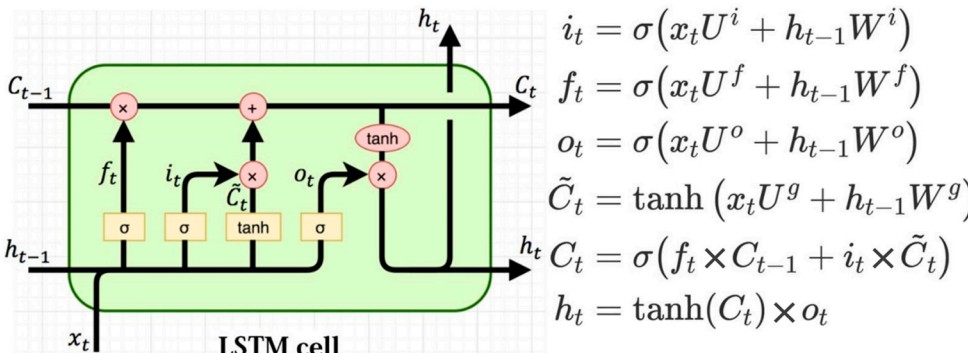

$$i_t = \sigma\big(x_t U^i + h_{t-1} W^i\big)$$

$$f_t = \sigma\big(x_t U^f + h_{t-1} W^f\big)$$

$$o_t = \sigma\big(x_t U^o + h_{t-1} W^o\big)$$

$$\tilde{C}_t = \tanh\big(x_t U^g + h_{t-1} W^g\big)$$

$$C_t = \sigma\big(f_t \times C_{t-1} + i_t \times \tilde{C}_t\big)$$

$$h_t = \tanh(C_t) \times o_t$$

**Figure 4.** Illustration of Long Short-Term Memory model; the gates equations illustrated on right-hand side.

Figure 5 uses a 3D scatter plot to visualize the LSTM model's analysis of SCADA network traffic over one day. Each point denotes an individual session, colored green if classified as normal or red if deemed threatening. Sessions are positioned along axes, indicating order, connected IP address, and predicted intrusion risk on a 0.0 to 1.0 scale. The prevalence of green points shows the model categorizing most of the traffic as normal. The red points reveal sessions flagged as potential threats. This color-coded scatter plot provides an intuitive demonstration of how the LSTM model monitors the larger flow of daily traffic to isolate anomalies and surface sessions meriting further investigation.

### 3.4. Technology Level—Network Segmentation (VLANs)

In OT network, IEEE 802.1Q VLANs were leveraged to divide the environment into separate security zones to isolate the most critical assets. Dedicated VLANs were established for each production line, as well as for sensitive systems like controllers and historians. This micro-segmentation contained threats by preventing lateral movement between areas if one was compromised.

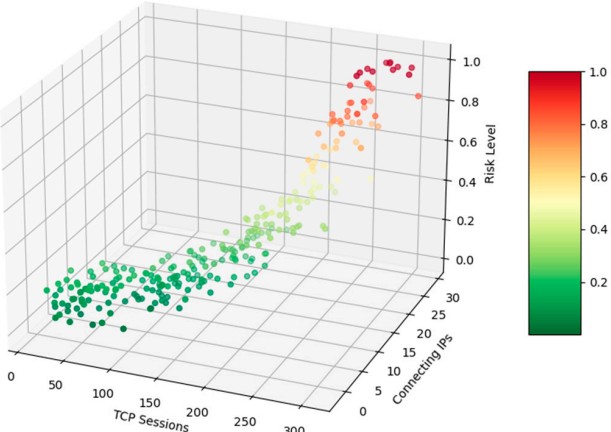

**Figure 5.** Three-dimensional scatter plot to visualize the LSTM model's analysis.

Inter-VLAN routing was carefully controlled via firewalls and gateways to allow only required connections. For example, the control center VLAN could access the historian VLAN but not directly interact with production floor endpoints. Cryptography was implemented to further secure necessary inter-VLAN communications, using IPsec VPNs between gateways.

By partitioning the OT network into isolated segments with restricted visibility, the attack surface was greatly reduced. VLANs provided the layer 2 separation that was needed, while additional controls at layers 3+ enabled secure routing between zones. This defense-in-depth approach limited potential infection blast radius while still permitting essential data flows. VLAN segmentation was found to be a key best practice, alongside proper network-level controls, for improving security posture in the SCADA environment.

## 4. Results

The implementation of a secure SCADA architecture has yielded significant measurable security improvements across key metrics over the 6 months following deployment. Unauthorized access events, as measured by the count of unauthorized access attempts per month, decreased by 57.4% from a monthly average of 3574 events down to 1523 events after introducing NFC badge authentication. This greatly reduced potential security breaches through strengthened access controls. The mean time to threat containment, calculated as the average elapsed time from threat detection to successful containment, saw a 41.2% reduction from 3.08 h down to 1.81 h due to newly implemented automated response capabilities. This enabled much faster reactions to and mitigation of identified threats. Hacking attempts against the fog-level MES dashboard, tracked via the count of hacking attempts in SIEM logs per week, declined sharply by 79.2% from a weekly average of 842 attempts down to 176 attempts following the hardening of the web interface through defenses like HTTPS and access tokens. The quantified enhancements across key security metrics like unauthorized access, threat containment time, and hacking attempts validate the efficacy of the secure SCADA implementation in improving the system's overall resilience against threats. Key gains were realized in reducing attack surfaces, enabling rapid automated threat response, and limiting potential breach impacts and blast radii.

Encrypting SCADA links completely prevented the tampering of sensor measurements, while VLAN micro-segmentation limited the potential blast radius damage. Together, these measures significantly strengthened the system's resilience against threats by reducing vulnerabilities. The measurable security advancements validate the effectiveness of the secure SCADA implementation in improving the system's overall security posture through reduced attack surfaces, rapid threat response, and limited breach impacts.

Figure 6 shows the charts depicting security improvements achieved through the implementation of a secure SCADA system across four key metrics. The most substantial improvement is seen in unauthorized access events, with a 57.4% reduction following the

introduction of NFC badge authentication. Automated threat response capabilities led to a 41.2% decrease in mean time to containment. In the SIEM logs, hacking attempts against the fog-level MES dashboard saw a sharp 79.2% decline after deploying defenses like HTTPS and access tokens. Finally, encrypting SCADA links led to 100% prevention of sensor measurement tampering.

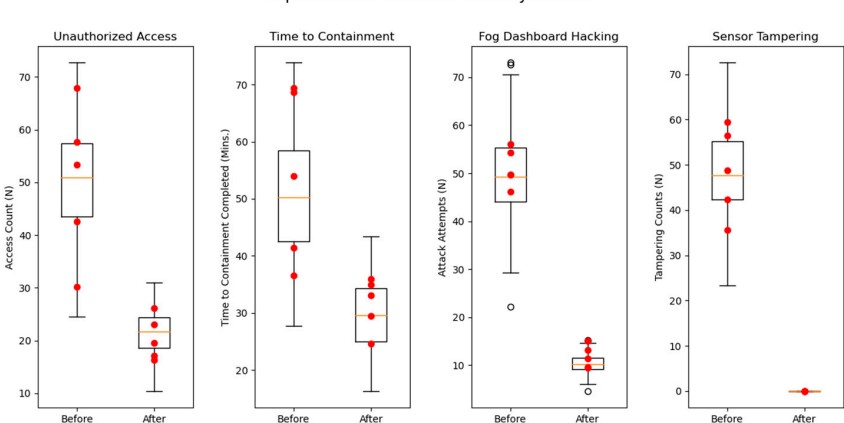

**Figure 6.** This figure shows the before and after values for 4 SCADA security metrics. The orange color lines are the medians and the rectangular boxes show the interquartile range (IQR). The values outside the boundaries (1.5 × IQR) are considered as outliers (hollow circles). The red dots are valid data points.

The deep learning-based IDS continuously monitors network activity for anomalous patterns indicative of potential intrusions. Upon such patterns exceeding defined threshold values of abnormality, the system automatically triggers alerts and initiates automated containment actions, such as blocking suspicious IP addresses or disabling associated user accounts. These capabilities enable rapid, programmatic responses to contain detected threats until more comprehensive forensic analysis and remediation can be performed. In the 6 months after the IDS was implemented, 17 intrusion events were detected. Moreover, 3 accounts were disabled, and 17 risky IP addresses were cut off.

Figure 7 shows intrusion detection system (IDS) activity over a typical production day. The standard shift runs from 08:00 to 20:00, with meal breaks at 12:00 and 18:00. SCADA bandwidth ranges up to 100 Mbps during production hours, dropping to a few Mbps after the production shift ends. The IDS continuously monitors network traffic for intrusion patterns, taking appropriate automated actions and sending alerts if configured detection thresholds are reached. In this study, the deep learning-based IDS detected anomalous intrusion patterns anytime throughout the day, regardless of production or meal hours.

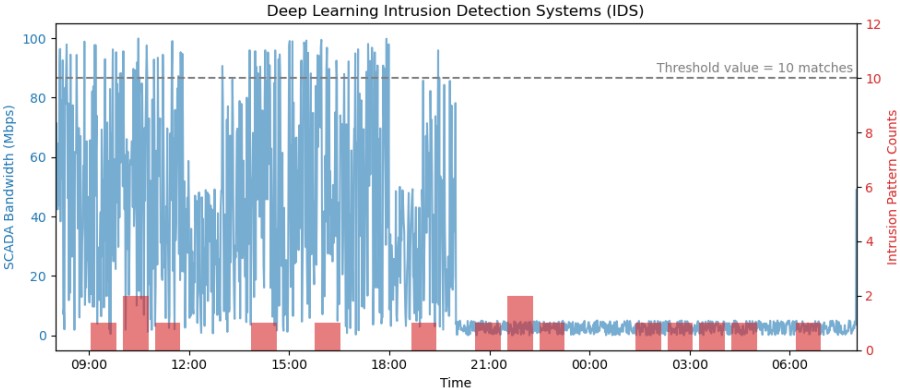

**Figure 7.** Intrusion detection system (IDS) activity. The red bars show the intrusion pattern detected counts and the blue line shows the minute peak bandwidth of SCADA activities.

As a corporate policy, deep learning IDS detected a total of 17 intrusion events over the 6 months following the implementation. As remediation actions, 3 accounts were disabled, and 17 suspicious IP connections were cut off.

## 5. Discussion

The quantified security improvements validate the effectiveness of implementing a comprehensive, multilayered cybersecurity strategy encompassing access controls, monitoring, encryption, and system hardening to protect SCADA environments. The integrated framework successfully reduced vulnerabilities across people, policies, processes, and technologies.

A key novelty of this approach is the multilayered integration of emerging capabilities like deep learning intrusion detection, granular access controls, and legacy system upgrading with traditional best practices like network segmentation and encrypted communications. This hybrid framework allows manufacturers to take advantage of new technologies while still leveraging established protections. The deep learning IDS, in particular, enables rapid automated threat detection and response, overcoming the limits of signature-based methods.

Another innovative aspect is the edge computing integration for access management. Pushing identity and access controls to the network edge enhances security and reduces dependence on the cloud. The lightweight mutual authentication policy also demonstrates securing legacy devices through updates like encrypted protocols rather than wholesale replacement.

This research highlights the importance of a proactive cybersecurity strategy as interconnectivity increases between OT and IT environments. As identified in the literature review, unprotected SCADA connectivity introduces significant risks that manufacturers must address. This framework provides a model to securely enable Industry 4.0 integration by mitigating vulnerabilities at multiple levels. Quantified metrics validated the threat detection, containment, and prevention capabilities achieved.

The multilayered methodology can help industrial organizations maintain resilient operations even as attacks grow more sophisticated. By combining the latest technologies with robust policies and processes, manufacturers can confidently pursue digital transformation initiatives with the knowledge that risks are managed. As automation and connectivity expand through concepts like smart factories and the IIoT, cybersecurity will only increase in importance. This research offers both an effective blueprint for SCADA protection and a foundation to build upon as technology evolves.

While this integrated framework demonstrated significant improvements in securing the SCADA environment, certain limitations should be noted. The costs associated with upgrading legacy hardware, implementing new access control systems, and developing customized deep learning algorithms may be prohibitive for some organizations. The complexity of managing multiple layered security controls could also pose implementation challenges. Further research is needed to streamline and simplify management across disparate tools. However, as demonstrated by the metrics, the risk-reduction benefits outweigh these limitations. Organizations must balance costs against the risks of cyber incidents in their unique environment.

## 6. Conclusions

This study presented a framework to secure SCADA systems in an electronics manufacturing facility using access controls, network segmentation, encrypted communications, deep learning intrusion detection, and legacy upgrades based on a multilayer approach encompassing people, policies, processes, and technologies. The defense-in-depth (DID) techniques strengthened edge and fog layer security to enable smart manufacturing. The proposed framework enhanced cybersecurity best practices while allowing more robust and efficient operation. The objective of empowering manufacturers to securely realize smart factory capabilities through a resilient SCADA architecture protected against emerging

risks was achieved and enabled electronics manufacturers to safely pursue Industry 4.0 integration.

For future research, exploring anomaly detection and automated response via digital twins could maintain cyber resilience as Industry 4.0 advances. By implementing complementary technical and administrative controls, this framework mitigates escalating threats from increased IT-OT interconnectivity. As automation and AI transform security capabilities, pursuing innovations in network modeling, policy automation, and proactive threat research represents a promising direction to secure industrial environments against next-generation threats.

**Author Contributions:** Conceptualization, E.W. and C.K.M.L.; methodology, E.W. and C.K.M.L.; implementation E.W., result and analysis, E.W. and C.K.M.L.; project supervision, E.W. All authors have read and agreed to the published version of the manuscript.

**Funding:** The project is funded by the Research Institute for Advanced Manufacturing of project code CD4E.

**Institutional Review Board Statement:** Not applicable.

**Informed Consent Statement:** Not applicable.

**Data Availability Statement:** The data presented in this article are available on request from the corresponding author.

**Acknowledgments:** The authors would like to thank the Research Institute of Advanced Manufacturing and VTech Communications Limited for their support of this study and their contribution to academic research in this area.

**Conflicts of Interest:** The authors declare no conflict of interest.

## Acronymous

| | |
|---|---|
| SCADA | Supervisory Control and Data Acquisition |
| CPPS | Cyber-Physical Production System |
| OT | Operational Technology |
| IT | Information Technology |
| IIoT | Industrial Internet of Things |
| IoT | Internet of Things |
| PLC | Programmable Logic Controller |
| RTU | Remote Terminal Unit |
| I/O | Input/Output |
| HMIs | Human–Machine Interfaces |
| BoM | Bill of Materials |
| COTS | Commercial Off-The-Shelf |
| ICS | Industrial Control Systems |
| DID | Defense-in-Depth |
| VPN | Virtual Private Network |
| IPsec | Internet Protocol Security |
| VLAN | Virtual Local Area Network |
| TLS | Transport Layer Security |
| HTTPS | Hypertext Transfer Protocol Secure |
| RF | Radio Frequency |
| ASK | Amplitude Shift Keying |
| MES | Manufacturing Execution System |
| SBC | Single Board Computer |
| NFC | Near Field Communication |
| IDS | Intrusion Detection System |
| SIEM | Security Information and Event Management |

| LSTM | Long Short-Term Memory |
|------|------------------------|
| RNN | Recurrent Neural Network |
| DL | Deep Learning |
| AI | Artificial Intelligence |

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
