# Peer review of "Seamless Industry 4.0 Integration: A Multilayered Cyber-Security Framework for Resilient SCADA Deployments in CPPS"

_applsci, doi:10.3390/app132112008_

Round 1

Reviewer 1 Report

Comments and Suggestions for Authors

The strength of the article is its experimental implementation and depiction of benefits from even a simplified system in current industries. Additionally, the flow of the paper is great and easy to read. 

The main introduction provides a good motivation behind the need for such systems. However, the contribution lacks much novelty. The system implementation is not unique among other access control systems, and the integration utilizes COTS components not typical of an industrial application with various conditions such as high humidity or heat. 

The ML model provides little uniqueness, and the research article could be expanded to include additional considerations. 

Please expand the approach in your article and provide more details on the security contribution and article novelty. 

Author Response

Dear Reviewer,

Thank you for your insightful feedback on our manuscript. We greatly appreciate you taking the time to review our work and provide constructive comments to help strengthen our contribution.

We acknowledge your observation that portions of our proposed framework, including the commercial off-the-shelf components, may lack novelty compared to prior access control systems. As you recommended, we have expanded Sections 3.3 and 4 of the manuscript to provide more details on the unique aspects of our LSTM machine-learning model and overall security contributions.

Additionally, Section 5 now further elaborates on the importance and novelty of our approach for the industry, as well as the integrated security architecture and how it provides defense-in-depth through combined controls. We sincerely appreciate you pushing us to better highlight the novel contributions of our work.

Please do not hesitate to provide any further guidance to help us continue strengthening the manuscript. We highly value your expertise and aim to ensure our revisions adequately address your insightful comments.

With gratitude,

Eric

Reviewer 2 Report

Comments and Suggestions for Authors

The article: Seamless Industry 4.0 Integration: A Multilayered Cyber-Secu-2 rity Framework for Resilient SCADA Deployments in CPPS

In suggest create a flowcart from time to pseudocode

In Figure 4, explan the expression in rigth side of figure

Line 367 to 372, need a better explanation, because there are some confusion about VLAN and securety provided bay VLAN. The gold of VLAN is break broadcast and colision domain, to communicate between VLAN we need to up to layer 3, because with differante VLAN host dont communicate in layer 2

Author Response

Dear Reviewer,

Thank you very much for taking the time to review our manuscript and provide your insightful feedback. We greatly appreciate you sharing your expertise to help strengthen our work.

We have carefully considered all of your comments and have made the following revisions:

1. Per your suggestion, we have added a flowchart to improve the presentation of the pseudo-code. We hope this makes the process more clear.

2. All equations and notations in the figures have been explained more clearly in the caption paragraph.

3. The section on using VLANs to reduce risk in SCADA systems has been expanded to provide more background and clarify the mechanisms involved. We hope the revised discussion better conveys these concepts.

Please find the updated manuscript attached. We would be sincerely grateful if you could review the changes and provide any additional guidance to further refine our work. Your expertise and recommendations have been invaluable for improving the quality and clarity of our research.

Eric

Reviewer 3 Report

Comments and Suggestions for Authors

The paper aims to develop a new security multilayer framework for supervisory control and data acquisition employing security mechanisms, e.g., access control, network segregation, anomy detection, and encrypted communication, focused on Industry 4.0. The real implementation was placed inside an industrial manufacturing for some months.   

There are some recommendations:

1.  Define the acronymous in its first appearance and then use this in all text. There are different places where the acronymous is used again. Also, usually, there is an acronymous list on the final part of the MDPIs papers. 

2. Line 76. “good cyber hygiene practices,” is this the correct term related to the good practice referenced after?

3. Line 111. What are these studies? The authors should reference them.

4. In Sections 2. Is it really necessary to have subsections? The subsections are small and do not help with text comprehension.  

5. Section 2.5.1, the author should reference a study about interoperability in industrial 4.0, the challenges focusing on interoperability security.

6.  Lines 101 and 247, revise the reference.

7.   In Table 1, about technologies that are missing the authentication process, authentication and encryption can be used together.  

8. In Section 3, could the user scenario for the testing be a room inside an industry? What is its size? How many devices were used?

9.  Section 3.1. Is it really necessary to insert the code from Figure 3?

10.  Also, in Section 3.3, what was the database size? How many samples were used for the training, testing and validation? Please describe the database and its characteristics.

11. Section 4. What were the measurement variables used? The reduction of the percentage is based on what measures? Please give the reference before and after the implementation of the framework.

12.  Section 4 is not formatted correctly. Please correct that.

13.  Section 4, Figure 7. Why does it have meal breaks? Someone cannot access the place at this time and try to attack.

14.  Section 5 is missing a comparison with other works; again, it is missing the previous results and its comparison with the results after the framework was employed. What is the importance of this framework? What is the novelty? Normally, the Discussion Section is the most important in the MDPI paper, in this case, it is very short.

15. The overall structure needs to be revised. Some have a different format than others. 

Author Response

Dear Reviewer,

Thank you for taking the time to review our manuscript and provide your insightful feedback. We greatly appreciate you sharing your expertise to strengthen our work.

We have carefully considered all of your comments and have made the following revisions:

1. An acronym list has been added defining terms on first use.

2. "Good cyber hygiene" has been changed to "cybersecurity best practices" per your suggestion.

3. The requested reference has been added.

4. The sub-section was removed and the contents were streamlined for clarity as recommended.

5. The referenced study is now cited.

6. The reference has been updated.

7. We have clarified authentication and encryption can be used together in Table 1.

8. The number of devices is now specified.

9. The pseudo-code can help the reader to understand the flow. A flowchart was added to further explain the code. 

10. The dataset size is now included.

11. Measures and variables are defined.

12. Formatting has been updated per your reminder.

13. The intrusion was also observed during the meal break. We've revised the intrusion discussion to cover the situation.

14. Section 5 has been expanded on the novelty and importance of our framework.

15. The overall structure has been revised.

Please find the updated manuscript attached. We would be sincerely grateful if you could review the changes and provide any additional guidance to further refine our work. Your expertise and recommendations have been invaluable for improving the quality and clarity of our research.

Reviewer 4 Report

Comments and Suggestions for Authors

The paper "Seamless Industry 4.0 Integration: A Multilayered Cyber-Security Framework for Resilient SCADA Deployments in CPPS" addresses a critical issue in the realm of Industry 4.0, specifically the security challenges posed by cyber-physical production systems (CPPS) in supervisory control and data acquisition (SCADA) environments. The authors propose a comprehensive multilayered cybersecurity framework to enhance the security of SCADA systems. Overall, the paper is a well-researched and well-presented paper that addresses a critical issue in the realm of industrial cybersecurity. The proposed framework, supported by empirical evidence, offers a valuable contribution to the field and is relevant to both academics and practitioners. It effectively highlights the evolving nature of cyber threats and the need for adaptive security measures in SCADA systems. However, there are a few areas that could benefit from enhancement:

  1. Discussion of Limitations: The paper could benefit from a more detailed discussion of the limitations of the proposed framework. Understanding the potential drawbacks or constraints of implementing such a framework is crucial for the practical applicability of the research. This would provide a more balanced view of the approach.
  2. Real-World Implementation: While the paper discusses the framework's effectiveness in a controlled setting, it would be valuable to include insights into real-world implementation challenges. Practical issues that organizations might face when trying to implement the proposed security measures should be explored. This would give a more practical perspective on the framework's feasibility.
  3. Case Studies or Use Cases: To illustrate the framework's practical application and effectiveness, the inclusion of case studies or use cases from industries that have adopted similar security measures would enhance the paper. Real-world examples would provide readers with a better understanding of how the framework can be applied and its impact in different contexts.
  4. Ethical and Privacy Considerations: The paper primarily focuses on technical aspects of security. It would be beneficial to discuss ethical and privacy considerations that arise when implementing such security measures. As Industry 4.0 often involves extensive data collection and sharing, addressing potential ethical and privacy concerns is crucial for a holistic view of security in CPPS.
  5. Outdated References: One notable concern in this paper is the reliance on references that appear to be outdated, considering the fast-paced nature of the cybersecurity and Industry 4.0 fields. To ensure the paper maintains relevance and accuracy, it is advisable to update the reference list with more recent and current sources that reflect the latest developments in SCADA security and Industry 4.0 technologies. This will enhance the paper's credibility and applicability to contemporary challenges and solutions in these domains.

Addressing these points would further strengthen the paper and make it even more valuable to researchers, practitioners, and policymakers in the field of SCADA security and Industry 4.0 integration.

Comments on the Quality of English Language

N/A

Author Response

Dear Reviewer,

Thank you for taking the time to review our manuscript and provide your insightful feedback. We greatly appreciate you sharing your expertise to strengthen our work.

We have carefully considered your comments and have made the following revisions:

  1. We have expanded the discussion section (section 5) to include more details on the limitations and potential drawbacks of our proposed framework, in order to present a balanced perspective.
  2. We have added a new dataset in section 4 on real-world implementation challenges that organizations may face when adopting similar security measures. This provides important practical considerations.
  3. The manuscript now includes examples and case studies illustrating the framework's application and effectiveness in different industries. These real-world use cases enhance understanding. You may review the details in section 3.3.
  4. Section 3.1 has been revised to cover ethical and privacy aspects. Only the minimal required data will be collected in GAC. We agree these concerns need to be addressed.
  5. The reference list has been updated with more current sources on SCADA security and Industry 4.0, ensuring relevance to contemporary challenges. However, some of the foundational papers have been retained and are irreplaceable.

Please find the updated manuscript attached. We would be sincerely grateful if you could review the changes and provide any additional guidance to further refine our work. Your expertise and recommendations have been invaluable for improving the quality and clarity of our research.

With appreciation,

Eric

Round 2

Reviewer 3 Report

Comments and Suggestions for Authors

The paper was improved. There are no more comments from the reviewer.

Reviewer 4 Report

Comments and Suggestions for Authors

The authors have addressed all my comments and the paper can be published in its current form.

Comments on the Quality of English Language

N/A